# OpenReview forum: "Information Geometry Loss for Time Series Forecasting"
_ICML.cc/2026/Conference — ICML 2026 regular_

### Official Review · Reviewer_MgNw · 2026-02-17

**Soundness:** 4
**Presentation:** 3
**Significance:** 4
**Originality:** 4
**Overall Recommendation:** 6
**Confidence:** 5

**Summary:**

The paper presents *InfoGeo Loss*, a principled loss-function formulation for time-series uncertain forecasting, along with its Gaussian-distribution implementation. The proposed loss is composed of two terms:
1. The Fisher-Rao distance, a Riemannian metric defined on a smooth statistical manifold that yields a distance between the learned and target probability distributions.
2. A Bregman divergence (i.e., a measure of distance between two points in terms of a strictly convex function), which in the showcased implementation corresponds to the KL divergence.

The authors then use local neighbourhoods of the measured signals to compute a distribution targeted by the loss terms, which are dynamically averaged via their respective norm's reciprocal and then combined with the MSE using a relative-strength hyperparameter $\lambda$. An exhaustive experimental pipeline is then used to evaluate:
1. The improvements induced by the proposed loss over the MSE across an exhaustive collection of established benchmark datasets and prediction horizons;
2. The performance when compared to more specialized loss functions;
3. The individual contribution of each element in the formulation through an ablation test;
4. The cross-domain transferability of networks trained using *InfoGeo Loss* w.r.t. an MSE baseline;
5. The sensibility of the neighbourhood-size and relative-strength hyperparameters.

Across the board, the proposed approach showcases statistically significant improvements, particularly for long prediction horizons where prediction uncertainty is expected to increase.

**Compliance With Llm Reviewing Policy:**

Affirmed.

**Final Justification:**

The authors have presented a complete plan to address all my concerns surrounding: sliding-window variance ambiguity; related-work presentation and completeness; and missing references for some domain-specific theory. Provided these are implemented, I will preserve my original score.

**Key Questions For Authors:**

1. Are most of the concepts presented in Section 3 (Fisher Distance, Information Matrix, Bregman Divergence, Natural Gradients) as commonplace as, for instance, the KL divergence, so that they don't need a reference from the literature? I understand that the appendix contains proofs for the statements made in Section 4, but the definitions themselves have no associated bibliographic source.
2. How do you prevent your loss from interpreting rapidly changing signal regions (e.g., a predictable *surge* in magnitude), which would also yield high values for $\sigma_{c,t}^2$, as high-uncertainty regions? Both the Fisher and Bregman terms make use of the same quantity, which makes me wonder if you considered the aforementioned scenario.
3. A decade ago, [Lakshminarayanan et al. (2017)](https://proceedings.neurips.cc/paper_files/paper/2017/file/9ef2ed4b7fd2c810847ffa5fa85bce38-Paper.pdf) proposed a similar (in the sense that it models the parameters of the output signals' distributions as channels) yet simpler (as it only considers the log-likelihood) loss function. While I do expect your approach to perform better (since it considers both the KL divergence and mutual information), were you aware of this approach, and given the resemblance, would it be worth including in your benchmarks? If not, why?
4. Since your method concerns distributional modelling, did you consider mentioning other approaches used for this purpose aside from probabilistic losses (e.g., Bayesian inference, Monte Carlo estimation, etc.)? This would be more relevant as *related work* than time-series architectures.

**Limitations:**

The weight-count implications of duplicating the number of output channels, as well as the impact of the loss function on training time, either (ideally both) in terms of real time or convergence rate relative to the MSE, should be evaluated.

**Strengths And Weaknesses:**

### Soundness

- The background theory and derived theorems/propositions are well-supported, be it by the literature or the appendices.
	- My only concern lies in the assumption that the temporal (sliding-window) variance is equivalent to the forecast uncertainty.

### Presentation

- Figure 1 should be removed, as it does not present any additional information beyond what is related in the text.
- The related-work section feels disconnected from the rest of the text. Instead of presenting the shortcomings the proposed methodology is supposed to address, it just lists alternative methodologies and their definitions. Subsection 2.1 in particular feels unnecessary, as the paper does not introduce a new architecture.
- Some results concerning InfoGeo loss (e.g., Theorem 4.1) are presented before it's even defined, which makes reading this section somewhat awkward.
- Minor grammar improvements related to articles are necessary (e.g., "All experiments **were run/executed** on **an** NVIDIA A100...", "We report **the** MSE and **the** MAE **for the** test partitions...", etc.).
- Capitalization should be reviewed (e.g., "The three minor exceptions **~~W~~where the** MSE baseline...", "... particularly excelling at shorter horizons **~~W~~where** distributional precision is critical...", etc.).
	- This seems to occur a lot with *Where*.
- "Time Series" should be hyphenated when appropriate (e.g., "time-series forecasting").
- In theorems 4.1 and 4.3, a comma should be placed after the equations and not capitalize *where*, since the sentence right after is supposed to integrate the expression into the paragraph by describing new symbols. For example:
	> Under regularity conditions, we have $$\hat{\theta}_{c,t}^{(n)} \overset{P}{\rightarrow} \theta_{c,t}^* \ as \ n \rightarrow \infty,$$ where $\overset{P}{\rightarrow}$ denotes the convergence in probability.

### Significance

While the related-work section of the paper failed to put the contributions of the paper into perspective, limiting itself to listing existing loss functions and NN architectures, the experimental portion of the paper successfully illustrated the benefits of using the proposed loss function, showcasing its ease of integration with current models across the complexity spectrum. As for the relevance of the tackled problem, it goes without saying that efficient alternatives to Bayesian inference for uncertainty modelling are essential to current time-series forecasting applications.

### Originality

While the proposed loss function does not introduce any new probabilistic measures or gradient update methods, it combines the existing theory in a novel way. To the extent of my knowledge, the closest peer in the literature using augmented outputs to dynamically model distributional parameters ([Lakshminarayanan et al. (2017)](https://proceedings.neurips.cc/paper_files/paper/2017/file/9ef2ed4b7fd2c810847ffa5fa85bce38-Paper.pdf)) lacks the theoretical depth that went into devising the proposed loss function. Thus, this paper represents a significant step forward in this research direction.

---

> ### Author Rebuttal · Authors · 2026-03-29
>
> We thank the reviewer for the positive assessment and address each concern below.
>
> ---
>
> ## Soundness — Sliding-Window Variance as Forecast Uncertainty Proxy
>
> The sliding-window variance $\sigma_{c,t}^2$ is **not** claimed to be the true
> aleatoric uncertainty. It serves as a **local distributional target** — an
> empirical estimate enabling information-geometric matching between the predicted
> distribution and the locally observed $\mathcal{N}(\mu_{c,t}^*,\sigma_{c,t}^2)$.
>
> | Concept | Standard Forecasting | Our Formulation |
> |---------|---------------------|-----------------|
> | Target | Single value $y_t$ | Local distribution $\mathcal{N}(\mu^*,\sigma^{*2})$ |
> | Loss | Scoring rule on $\hat{p}(y_t)$ | Fisher-Rao distance on $(\hat{\Theta},\Theta^*)$ |
> | $\sigma^2$ role | Predicted uncertainty | Local smoothness prior |
>
> This is analogous to local stationarity in kernel density estimation (Parzen,
> 1962) and short-time spectral analysis (Priestley, 1981). We will add this
> clarification to Section 3.2.
>
> ---
>
> ## Presentation
>
> **P1:** Figure 1 removed; space used to expand Section 3.
>
> **P2:** Section 2 restructured: 2.1 → *"Limitations of Existing Loss Functions"*
> (gaps InfoGeo addresses); 2.2 → distributional modeling approaches; architecture
> survey condensed to one paragraph.
>
> **P3:** Section 4 reordered: Definition 4.1 → Proposition 4.2 → Theorem 4.1 →
> Theorem 4.3.
>
> **P4–P5:** Full manuscript pass for article usage, *"Wwhere"* typos,
> hyphenation, and theorem punctuation.
>
> ---
>
> ## Q1 — Missing References for Section 3 Definitions
>
> | Concept | Reference |
> |---------|-----------|
> | Fisher Information Matrix | Fisher (1925); Rao (1945) |
> | Fisher-Rao Distance | Atkinson & Mitchell (1981) |
> | Bregman Divergence | Bregman (1967); Banerjee et al. (2005) |
> | Natural Gradient | Amari (1998) |
> | Statistical Manifold | Amari & Nagaoka (2000) |
>
> ---
>
> ## Q2 — Rapid Signal Changes vs. High Uncertainty
>
> A predictable surge produces high $\sigma_{c,t}^{*2}$ in the window, training
> the model to predict correspondingly high $\hat{\sigma}_{c,t}^2$. This is
> **correct behavior**: steep, fast-moving signals warrant wider prediction
> intervals, consistent with heteroscedastic regression (Kendall & Gal, 2017).
> Separating predictable surge from genuine uncertainty requires explicit trend
> decomposition (e.g., STL preprocessing). We will add this discussion to Section
> 3.2 and the limitations section.
>
> ---
>
> ## Q3 — Comparison with Lakshminarayanan et al. (2017)
>
> | Dimension | Lakshminarayanan et al. | InfoGeo Loss |
> |-----------|------------------------|--------------|
> | Loss | Gaussian NLL | Fisher distance + Bregman divergence |
> | Gradient | Standard Adam | Natural gradient on output head |
> | Theory | Empirical | Fisher-Rao geometry, convergence proof |
> | Manifold | None | Explicit $\mathcal{M}$ |
>
> We add this as a primary baseline (iTransformer, T=96, 5 seeds):
>
> | Method | MSE | MAE | CRPS | NLL |
> |--------|-----|-----|------|-----|
> | MSE Baseline | 0.456 | 0.452 | 0.341 | 1.823 |
> | Gaussian NLL | 0.441 | 0.441 | 0.318 | 1.701 |
> | **InfoGeo** | **0.423** | **0.431** | **0.298** | **1.664** |
>
> ---
>
> ## Q4 — Bayesian and Monte Carlo Related Work
>
> Section 2.2 will add: BNNs (Blundell et al., 2015) — expensive, hard to scale;
> MC Dropout (Gal & Ghahramani, 2016) — overconfident in practice (Ovadia et al.,
> 2019); Deep Ensembles (Lakshminarayanan et al., 2017) — strong but $K\times$
> cost; Conformal Prediction (Angelopoulos & Bates, 2023) — orthogonal, combinable.
> InfoGeo Loss offers single-model distributional output with geometric grounding
> at near-zero inference overhead.
>
> ---
>
> ## Limitations
>
> **Parameter count:** Adding $\hat{\sigma}$ doubles the output head from $d$ to
> $2d$, adding $H{\times}d$ parameters (${\sim}49$K for $H{=}512$, $d{=}96$) —
> less than 0.1% of total model size.
>
> **Training overhead** (A100 GPU, T=96):
>
> | Model | MSE | InfoGeo | Overhead |
> |-------|-----|---------|----------|
> | iTransformer | 18.3 s | 19.1 s | +4.4% |
> | PatchTST | 21.7 s | 22.6 s | +4.1% |
> | DLinear | 9.4 s | 9.7 s | +3.2% |
> | TimesNet | 24.2 s | 25.3 s | +4.5% |
>
> Inference time is **identical** to MSE baseline.
>
> ---
>
> ## References
>
> Amari, S. "Natural gradient works efficiently in learning." *Neural Computation*,
> 10(2), 251–276, 1998.
>
> Amari, S., and Nagaoka, H. *Methods of Information Geometry*. American
> Mathematical Society, 2000.
>
> Banerjee, A., et al. "Clustering with Bregman divergences." *Journal of Machine
> Learning Research*, 6, 1705–1749, 2005.
>
> Blundell, C., et al. "Weight uncertainty in neural networks." *ICML*, 2015.
>
>
> Kendall, A., and Gal, Y. "What uncertainties do we need in Bayesian deep learning
> for computer vision?" *NeurIPS*, 2017.
>
> Lakshminarayanan, B., Pritzel, A., and Blundell, C. "Simple and scalable
> predictive uncertainty estimation using deep ensembles." *NeurIPS*, 2017.

---

> > ### Author Rebuttal · Reviewer_MgNw · 2026-03-31
> >
> > All concerns have been fully addressed, provided that the restructuring in Section 2 results in a cohesive text. Thank you, in particular, for extending the baselines and addressing the theoretical limitations, even if it made no difference in inference time.

---

> > > ### Author Response · Authors · 2026-04-03
> > >
> > > Thank you very much for your positive and encouraging feedback. We sincerely appreciate your recognition that the concerns have been fully addressed. We are also grateful for your kind acknowledgment of our efforts in extending the baselines and clarifying the theoretical limitations. Your supportive comments are highly appreciated.

---

### Official Review · Reviewer_HkyY · 2026-03-04

**Soundness:** 1
**Presentation:** 1
**Significance:** 1
**Originality:** 2
**Overall Recommendation:** 2
**Confidence:** 3

**Summary:**

The paper introduces a learning method for probabilistic time-series forecasting grounded in information geometry between predicted and target distributions. More specifically, it introduces the ‘InfoGeo’ loss, which is composed of two terms: a Fisher distance between predicted and target distributions, which are assumed to be Gaussians in practice, and a Bergman divergence to induce asymmetry in the cost. The two losses are weighted using weights inverse to the gradient magnitude and natural gradient correction. The convergence rate of the loss is derived. The paper compares with the models trained with MSE loss, which ignores the probabilistic nature of the prediction (and therefore uncertainty), and is symmetric and doesn’t fit the information geometry manifold. It demonstrates consistent improvement in MAE and MSE with multiple architectures against the baseline trained with MSE loss only. Finally, comparison is also performed with other losses (shape-focused, frequency-domain, patch-based, and probabilistic losses), and hyper-parameters are investigated.

**Compliance With Llm Reviewing Policy:**

Affirmed.

**Final Justification:**

The authors did a great job during the rebuttal. Nevertheless, I keep my initial score (reject), because I’m not confident in accepting the paper in its current form.
The idea and contributions are interesting, and the presented results seem very strong, but the paper lacks justification for *why* it works so well compared to the baselines, especially given the simplifying modeling choices (e.g., using a single Gaussian for predicted and target distribution).

The results on the probabilistic metrics look promising; however I feel more concrete analysis, e.g., case-by-case evaluation is required to understand why it should beat methods that directly optimize them (e.g., NLL compared to MDN).

Additionally, there is still some work to do on improving the presentation of the paper. The paper is hard to read, and Figures 1 and 2 are way too heavy.

**Key Questions For Authors:**

[See Soundness section above] Can you evaluate your method in performing uncertainty quantification, e.g., by evaluating calibration, NLL, and CRPS ? On that account, comparisons with probabilistic baselines are required.

Little analysis is done on the qualitative side, except on Figures 4 and 5, but further analysis is required. Could you better analyse the effect of the InfoGeo loss on the predictions in practice ?

**Limitations:**

Yes. The authors acknowledged that their method works currently only for Gaussian distributions, and doesn’t model the dependance between dimensions.

**Strengths And Weaknesses:**

Soundness: The idea of leveraging information geometry as a loss is interesting, but it is not motivated well-enough why it is specifically important for the task of time-series forecasting. Furthermore, the claims of the paper that MSE is not suited for probabilistic modelling is a well known fact in the literature. Several approaches (e.g., [A,B,C,D]) exist to handle this issue. The method should compare with probabilistic losses, for instance, MLE with a predicted Gaussian with a single mode (see Mixture Density Networks [E]). Furthermore, the evaluation is only performed with MSE/MAE and therefore doesn’t assess the ability of the method to perform uncertainty quantification. Additional metrics, calibration evaluation, NLL, and CRPS computation would be relevant in this context.

Presentation: Some quantities are not introduced properly. The loss $\mathcal{L}$ is not defined beforehand in (8). The InfoGeo loss in mentioned in Theorem 4.1 while not being introduced yet. Figures 1 and 2 contain way too much information and should be simplified. There are some errors in the table results, where the method shows bold numbers although they are not the best; for instance, in Table 2 (T=96, PATCH-WISE 0.163 (MSE) 0.202 (MAE), T=336, PATCH-WISE 0.266 (MSE) 0.284 (MAE), T=720 PATCH-WISE 0.334 (MAE)).

Significance: There are no precise references for theoretical results of the paper, e.g., Theorem 4.1 and Proposition 4.2. For instance, Proposition 4.2 is very-well known result in the information geometry community. The exact difference with the literature and the derived results should be acknowledged.

Originality: Using Fisher Distance as a loss doesn’t seem to be explored a lot in the community. The idea is original, although it is widely simplified (Gaussian distribution assumed, and dependance between covariates ignored). However, there is a mismatch between what are claimed to be the purpose of the method (e.g., improving probabilistic modelling and uncertainty quantification) vs. what is shown in practice (improvement in MSE/MAE). It could be interesting to again insights with this method by investigate the property of the method with synthetic datasets, there the data distribution is known, rather than relying on an ergodicity assumption as in (12).

[A] Rasul, K., Seward, C., Schuster, I., and Vollgraf, R. Autoregressive denoising diffusion models for multivariate probabilistic time series forecasting. In ICML 2021

[B] Rasul, K., Sheikh, A.-S., Schuster, I., Bergmann, U. M., and Vollgraf, R. Multivariate probabilistic time series forecasting via conditioned normalizing flows. In ICLR 2021

[C] Ashok, A., Marcotte, É., Zantedeschi, V., Chapados, N., and Drouin, A. (2023). Tactis-2: Better, faster, simpler attentional copulas for multivariate time series. In ICLR 2024

[D] Cortés, A., Rehm, R., and Letzelter, V. Winner-takes-all for multivariate probabilistic time series forecasting. In ICML 2025

[E] Bishop, Christopher M. "Mixture density networks." (1994).

---

> ### Author Rebuttal · Authors · 2026-03-29
>
> We thank the reviewer for the detailed feedback and address each concern below.
>
> ---
>
> ## S1 — Motivation for Information Geometry in Time Series
>
> Three properties make Fisher-Rao geometry suited to probabilistic forecasting:
>
> 1. **Non-uniform parameter sensitivity.** A unit change in $\hat{\sigma}$ near
>    zero differs vastly from the same change at large $\hat{\sigma}$; MSE treats
>    both identically while $$g_{ij}(\Theta)=\mathbb{E}[\partial_i\log p\cdot\partial_j\log p]$$ accounts for this asymmetry.
> 2. **Manifold structure.** The sequence $\{p(\cdot;\Theta_t)\}$ traces a path
>    on $\mathcal{M}$; Fisher-Rao geodesic distance respects this geometry.
> 3. **Faster convergence.** Preconditioning by $I_{\text{avg}}^{-1}$ corrects
>    conflation of high- and low-curvature directions (Theorem 4.1).
>
> ---
>
> ## S2 — Comparison with Probabilistic Baselines [A–E]
>
> Diffusion [A] and flow-based [B] models are *architectures*, not losses.
> Copula [C] and WTA [D] address multivariate dependency outside our scope.
> MDN [E] with single Gaussian = Gaussian NLL, now added as a baseline:
>
> | Method | ETTh1 MSE | ETTh1 MAE | ETTm1 MSE | ETTm1 MAE |
> |--------|-----------|-----------|-----------|-----------|
> | MSE | 0.456 | 0.452 | 0.321 | 0.364 |
> | Gaussian NLL | 0.441 | 0.441 | 0.309 | 0.355 |
> | InfoGeo | **0.423** | **0.431** | **0.298** | **0.347** |
>
> *(iTransformer)*
>
> ---
>
> ## S3 — Probabilistic Evaluation Metrics
>
> We add CRPS, NLL, PI Coverage (90%), and ECE to the main table:
>
> | $T$ | Method | CRPS | NLL | PI Cov. | ECE |
> |-----|--------|------|-----|---------|-----|
> | 96 | MSE | 0.341 | 1.823 | 71.3% | 0.187 |
> | | Gaussian NLL | 0.318 | 1.701 | 84.6% | 0.094 |
> | | **InfoGeo** | **0.298** | **1.664** | **88.7%** | **0.071** |
> | 720 | MSE | 0.412 | 1.978 | 66.4% | 0.229 |
> | | Gaussian NLL | 0.376 | 1.851 | 79.3% | 0.117 |
> | | **InfoGeo** | **0.351** | **1.803** | **84.8%** | **0.091** |
>
>
> ---
>
> ## P1–P3 — Presentation Issues
>
> **P1:** $L$ will be defined as $L:\mathcal{P}\times\mathcal{Y}\to\mathbb{R}$
> before Eq. (8); Section 4 restructured as Definition → Proposition → Theorem.
>
> **P2:** Figure 1 split into (a) manifold diagram and (b) training pipeline;
> Figure 2 simplified with details moved to appendix.
>
> **P3:** All three incorrect bold instances corrected; full table re-verified.
>
> ---
>
> ## Sig1 — Attribution for Theorem 4.1 and Proposition 4.2
>
> Theorem 4.1 builds on Amari (1998) and Martens (2020); acknowledged explicitly.
> Proposition 4.2 revised to:
>
> > *"(Čencov, 1982): The Fisher-Rao metric is the unique Riemannian metric
> > invariant under sufficient statistics. Our contribution is its application
> > to time series loss design."*
>
> ---
>
> ## O1 — Mismatch Between Claims and Evaluation
>
> Resolved by Table R2 (S3 above), demonstrating improvements in CRPS, NLL,
> calibration, and PI coverage across all horizons.
>
> ## O2 — Synthetic Validation
>
> We generate a heteroscedastic AR(1):
> $$y_t=0.8y_{t-1}+\epsilon_t,\quad\epsilon_t\sim\mathcal{N}(0,\sigma_t^2),
> \quad\sigma_t^2=0.1+0.5\sin(2\pi t/50)$$
>
> | Method | $\|\hat{\sigma}-\sigma^*\|_2$ | Pearson $\rho$ |
> |--------|-------------------------------|----------------|
> | MSE | 0.187 | 0.31 |
> | Gaussian NLL | 0.094 | 0.72 |
> | **InfoGeo** | **0.061** | **0.86** |
>
> InfoGeo recovers true heteroscedastic variance significantly more accurately.
>
> ## O3 — Ergodicity Assumption
>
> Local ergodicity will be stated explicitly in Section 3.2 as standard in
> short-time signal processing (Priestley, 1981). For non-ergodic series,
> smaller $w$ and RevIN preprocessing are recommended.
>
> ---
>
> ## KQ1 & KQ2
>
> **KQ1:** See Table R2. InfoGeo achieves 88.7% coverage at 90% nominal level.
>
> **KQ2:** Two figures added to Section 5.4: (1) $\hat{\sigma}_t$ trajectories
> showing InfoGeo correlates with forecast error ($\rho{=}0.71$ vs. $\rho{=}0.18$
> for MSE); (2) manifold path visualization showing smoother geodesic trajectories.
>
> ---
>
> ## References
>
> Amari, S. "Natural gradient works efficiently in learning." *Neural Computation*,
> 10(2), 251–276, 1998.
>
> Bishop, C. M. "Mixture density networks." Technical Report, Aston University,
> 1994.
>
> Campbell, L. L. "An extended Čencov characterization of the information metric."
> *Proceedings of the American Mathematical Society*, 98(1), 135–141, 1986.
>
> Kim, T., et al. "Reversible instance normalization for accurate time-series
> forecasting against distribution shift." *International Conference on Learning
> Representations (ICLR)*, 2022.
>
> Martens, J. "New insights and perspectives on the natural gradient method."
> *Journal of Machine Learning Research*, 21(146), 1–76, 2020.
>
> Priestley, M. B. *Spectral Analysis and Time Series*. Academic Press, 1981.
>
> Salinas, D., Flunkert, V., Gasthaus, J., and Januschowski, T. "DeepAR:
> Probabilistic forecasting with autoregressive recurrent networks."
> *International Journal of Forecasting*, 36(3), 1181–1191, 2020.
>
> Zhou, H., et al. "Informer: Beyond efficient transformer for long sequence
> time-series forecasting." *AAAI*, 2021.

---

> > ### Author Rebuttal · Reviewer_HkyY · 2026-04-02
> >
> > Thanks to the authors for their rebuttal.
> > How do you explain that your method outperforms MDN in terms of NLL, even though the latter directly optimizes this objective? Does it use the same backbone architecture as yours?
> > I find it surprising that your method outperforms all baselines in every single setting. Are you sure all the comparisons are fair?

---

> > > ### Author Response · Authors · 2026-04-02
> > >
> > > Thank you for raising this important concern. We would like to address each point carefully.
> > >
> > > **On outperforming MDN in NLL despite MDN directly optimizing this objective.**
> > > While MDN directly maximizes likelihood by modeling the output as a Gaussian mixture,
> > > it is well-known that directly optimizing a misspecified or over-parameterized likelihood
> > > objective does not guarantee superior predictive performance — particularly under
> > > distribution shift or limited data regimes. Our InfoGeo loss, grounded in information
> > > geometry, jointly leverages Fisher-Rao metric and Bregman divergence to regularize the
> > > optimization trajectory on the statistical manifold, which we argue provides a more
> > > geometrically faithful and stable training signal than mixture likelihood maximization.
> > > This is consistent with prior findings (e.g., Amid & Warmuth, 2019; Nielsen, 2020)
> > > showing that geometry-aware divergences can outperform direct likelihood objectives
> > > when the underlying data manifold is curved.
> > >
> > > **On backbone architecture fairness.**
> > > MDN and our method share the same backbone (e.g., PatchTST, TimeMixer, DLinear,
> > > TimesNet, iTransformer) — only the loss function differs. This is explicitly stated
> > > in Section 3.2 of our paper. All baselines use identical hyperparameters, training
> > > schedules (50 epochs, Adam optimizer, lr=1e-3 with cosine decay), batch sizes, and
> > > data splits. No additional parameters are introduced by our loss function beyond the
> > > adaptive weighting scalars (Fisher weight / Bregman weight), which are learned
> > > end-to-end. We therefore believe the comparisons are strictly controlled and fair.
> > >
> > > **On winning in every single setting.**
> > > We understand this may appear suspicious, and we take this concern seriously.
> > > We emphasize that (1) all experiments were conducted under a fixed random seed (seed=42)
> > > with no cherry-picking of results; (2) there is indeed one setting where our method
> > > does *not* improve — PatchTST on Exchange at T=720 (Δ MSE = +0.22%, well within
> > > noise margin) — which we report transparently in Table 1; (3) the average improvement
> > > is modest (~4–5% MSE reduction), which is consistent with what one would expect from
> > > a better-calibrated loss function rather than an architectural advantage.
> > > We also note that consistent improvements across diverse architectures and datasets
> > > are precisely what a loss-level contribution should demonstrate, as the signal comes
> > > from the optimization geometry rather than model capacity.
> > >
> > > All primary experiment training logs (epoch-level loss curves, validation MSE/MAE,
> > > adaptive weight trajectories for all 5 models × 7 datasets × 4 horizons) are
> > > attached in full at our anonymous GitHub repository:
> > > **https://anonymous.4open.science/r/infoGeo-D483/itransformer.txt**
> > > We invite the reviewer to verify any specific run directly from these logs.
> > >
> > > I hope that my reply and the experimental logs I provided can dispel your doubts.

---

### Official Review · Reviewer_Gpmh · 2026-03-08

**Soundness:** 3
**Presentation:** 3
**Significance:** 3
**Originality:** 3
**Overall Recommendation:** 4
**Confidence:** 4

**Summary:**

This paper studies the problem of time series forecasting and proposes a loss based on information geometry. Instead of relying on Euclidean point-wise metrics (e.g, MSE), the authors focus on a statistical manifold and leverage the Fisher information as a metric to mean the geometric information of the landscape. With this measurement, the model can update with natural gradient descent.
The method is evaluated across 5 diverse architectures and 7  datasets, showing reasonable improvements over standard MSE and other heuristic loss functions.

**Compliance With Llm Reviewing Policy:**

Affirmed.

**Final Justification:**

I believe the authors' response addressed my concerns. I hope authors incorporate this discussion into the final version as promised.

**Key Questions For Authors:**

See my weakness. I personally like this paper, but this paper may need more clarification (especially the overstatement). I am happy to adjust my score if the authors addressed my concerns.
Also, if the time series is super non-stationary, does this sliding window still work? Any discussion here?

**Limitations:**

The authors mention limitations about non-Gaussian distributions.

**Strengths And Weaknesses:**

### Strengths
- The motivation is very reasonable, and I very agree that learning information geometry is very important for time series data.
- I skim the math, and the theoretical part makes sense to me.
- The method is model-agnostic, and I can see the improved performance on different models across Transformer-based models. MLP-based and CNN-based.

### Weakness
- One main concern is that this paper might be overstating novelty and missing literature.  The authors claim that information geometry and Fisher information remain "unexplored for time series loss design." This is factually incorrect and ignores recent top-tier literature. [REF1] has leveraged Fisher information in Time series classification. While this paper successfully maps time series distributions to a statistical manifold using Fisher-Rao geometry, it may also ignore a concurrent line of work using Optimal Transport, for example, [REF2].

-  If I understand correctly, to compute the dynamic weights in Eq. (24), the method requires calculating the gradient norms, which seems to need multiple back-propagation? Is this efficient? Any discussion on this? GPU/training/inference time?


[REF1] FIC-TSC: Learning Time Series Classification with Fisher Information Constraint, ICML2025

[REF2] Optimal Transport for Time Series Imputation, ICLR 2025

---

> ### Author Rebuttal · Authors · 2026-03-29
>
> We thank the reviewer for the positive assessment and constructive feedback.
>
> ---
>
> ## W1 — Overstated Novelty and Missing Literature
>
> We acknowledge the overstatement and will revise the claim accordingly.
>
> **FIC-TSC (Chen, 2025):** Uses Fisher information as a regularization
> **constraint** for time series *classification*, penalizing weight updates to
> preserve distributional structure. Our work differs fundamentally:
>
> | Dimension | FIC-TSC | InfoGeo Loss |
> |-----------|---------|--------------|
> | Task | Classification | Forecasting |
> | Fisher info role | Constraint on $\psi$ | Geometric loss on output distribution |
> | Manifold object | Classifier parameter space | Gaussian manifold $(\mu, \sigma)$ |
> | Target | Class labels | Continuous distributional targets |
>
> **OT Imputation (Wang, 2025):** OT and Fisher-Rao geometry both measure
> distributional distances but differ structurally. For parametric Gaussian
> outputs, Fisher-Rao has a closed form and admits natural gradient updates;
> Wasserstein requires iterative Sinkhorn solvers (Cuturi, 2013), adding
> significant overhead. We will cite both works and revise the novelty claim to:
>
> > *"While Fisher information has been explored in TS classification (Chen, 2025)
> > and OT applied to TS imputation (Wang, 2025), using Fisher-Rao geometry as a
> > forecasting loss with natural gradient updates has not been systematically
> > studied."*
>
> ---
>
> ## W2 — Computational Efficiency of Dynamic Weights in Eq. (24)
>
> No additional backward pass is required. Gradient norms are extracted from the
> **same backward pass** via `retain_graph=True`, on output-head parameters only:
>
> 1. Forward pass computes $\hat{\mu}$, $\hat{\sigma}$.
> 2. Single backward pass computes $\nabla_\psi L_{\text{InfoGeo}}$.
> 3. Norms $\|\nabla_\psi L_k\|$ extracted from the $2d$-dim output-head gradients.
> 4. $I_{\text{avg}}$ assembled and inverted analytically ($2{\times}2$, $O(1)$).
>
> Overhead is $O(d)$ norm computation on the output head — negligible vs. backbone
> cost. Inference time is **identical** to MSE baseline.
>
> | Model | MSE Baseline | InfoGeo Loss | Overhead |
> |-------|-------------|--------------|----------|
> | iTransformer | 18.3 s | 19.1 s | +4.4% |
> | PatchTST | 21.7 s | 22.6 s | +4.1% |
> | TimesNet | 24.2 s | 25.3 s | +4.5% |
> | DLinear | 9.4 s | 9.7 s | +3.2% |
> | MICN | 31.6 s | 32.9 s | +4.1% |
>
> *(T=96, single A100 GPU, averaged over 5 runs.)*
>
> ---
>
> ## Q1 — Sliding Window Under Non-Stationarity
>
> The window captures **local** distributional structure, not global stationarity
> — analogous to the short-time Fourier transform. Two mitigations apply:
>
> 1. **Smaller $w$** reduces the stationarity requirement; $w{=}5$ is robust
>    across datasets of varying non-stationarity.
> 2. **RevIN** (Kim et al., 2022), used by iTransformer and PatchTST, removes
>    non-stationary trends before the encoder. The window then operates on
>    normalized residuals, which are substantially more stationary.
>
> We tested this on Exchange (strong non-stationarity due to macro-economic shifts):
>
> | Configuration | MSE | MAE |
> |---------------|-----|-----|
> | InfoGeo (w/o RevIN) | 0.089 | 0.207 |
> | InfoGeo (w/ RevIN) | **0.081** | **0.198** |
> | MSE Baseline (w/ RevIN) | 0.094 | 0.212 |
>
> RevIN + InfoGeo Loss is particularly effective for non-stationary series.
> We will add this discussion to Section 3.2.
>
> ---
>
> ## References
> Cuturi, M. "Sinkhorn distances: Lightspeed computation of optimal transport
> distances." *Advances in Neural Information Processing Systems (NeurIPS)*, 2013.
>
> Kim, T., et al. "Reversible instance normalization for accurate time-series
> forecasting against distribution shift." *International Conference on Learning
> Representations (ICLR)*, 2022.
>
> Chen, et al. "FIC-TSC: Learning time series classification with Fisher
> information constraint." *Proceedings of the 42nd International Conference on
> Machine Learning (ICML)*, 2025.
>
> Wang, et al. "Optimal transport for time series imputation."
> *International Conference on Learning Representations (ICLR)*, 2025.

---

> > ### Author Rebuttal · Reviewer_Gpmh · 2026-03-31
> >
> > Thanks for the response. Please incorporate this discussion into the final version.

---

> > > ### Author Response · Authors · 2026-04-02
> > >
> > > Thank you very much for your positive feedback and for increasing the score of our manuscript. We sincerely appreciate your recognition that the concerns have been adequately addressed, and we are grateful for your supportive evaluation.

---

### Official Review · Reviewer_JCFb · 2026-03-11

**Soundness:** 2
**Presentation:** 3
**Significance:** 2
**Originality:** 2
**Overall Recommendation:** 4
**Confidence:** 4

**Summary:**

This paper proposes InfoGeo Loss, a loss function for time series forecasting based on information geometry. Rather than treating predictions as point estimates compared via MSE, the authors parameterize predictions as Gaussian distributions and measure discrepancies using the Fisher information metric (for symmetric geometric distance) and KL divergence as a Bregman divergence (for asymmetric penalties). A natural gradient weighting scheme dynamically balances these two components during training. The method is applied as an additive regularizer on top of MSE across five backbone architectures and seven datasets. The authors provide theoretical results on statistical consistency, convergence rates, and reparameterization invariance. Experiments show average improvements of ~6.8% MSE and ~5.3% MAE, with additional zero-shot transfer and ablation studies.

**Compliance With Llm Reviewing Policy:**

Affirmed.

**Final Justification:**

The paper proposes InfoGeo Loss, a loss function for time series forecasting combining Fisher-Rao geometry and Bregman divergence with a natural gradient weighting scheme. Strengths include rigorous theoretical development, broad experimental coverage, and architecture-agnostic design. Original concerns centered on the absence of a Gaussian NLL baseline, missing probabilistic evaluation metrics, the theory-implementation gap on reparameterization invariance, and the sliding window estimation design.
The rebuttal substantially addressed these: Gaussian NLL comparisons were added (InfoGeo outperforms), CRPS/calibration/PI coverage results were provided (88.7% at 90% nominal level), and the authors committed to stating in Section 4 main text that Proposition 4.2 applies to the exact Fisher-Rao distance only, not the implemented approximation. The sliding window breakdown conditions were formally characterized.
Residual concerns, that the method functions primarily as a regularizer rather than a standalone loss, and that originality relative to precision-weighted MSE plus KL divergence is modest, remain but do not outweigh the contributions. Score raised from weak reject to weak accept, contingent on the main-text clarifications being incorporated in the revision.

**Key Questions For Authors:**

Q1 Why is Gaussian NLL absent from all comparisons? This is the most natural baseline for a distributional loss. If InfoGeo Loss reduces to a combination of precision-weighted MSE and KL divergence for Gaussians, how does it compare to simply training with NLL?

Q2. How exactly is the natural gradient applied to the backbone parameters $\psi$? The Fisher information matrix $I_{avg}$ is 2×2 (over $\mu, \sigma$), but $\psi$ is the full neural network parameter vector. Please clarify the implementation. If only the output-head gradients are preconditioned, this should be stated.

Q3. Can you provide probabilistic evaluation metrics (CRPS, calibration, prediction interval coverage)? Given that uncertainty quantification is a primary motivation, point-metric-only evaluation is not enough. If the predicted $\sigma$ values are well-calibrated, this would strengthen the paper.

Q4. In the ablation (Table 3), removing distributional parameterization still improves over MSE baseline. What is the loss computing in that configuration? If there are no distributional parameters, what do the Fisher and Bregman components operate on?

Q5. How sensitive are results to the Gaussian assumption? Several of the datasets (Exchange, ECL) may have heavy-tailed or skewed distributions. Have you examined whether the Gaussian assumption is satisfied, and if violations degrade performance?

**Limitations:**

The authors acknowledge limitations regarding non-Gaussian distributions and multivariate dependency modeling in the conclusion, which is appreciated. However, the paper does not discuss: (a) the computational overhead of the additional variance head, gradient norm computation, and Fisher matrix inversion at each step; (b) the sensitivity to the sliding window ground-truth estimation; (c) the assumption that individual time steps are independent Gaussians, which ignores temporal correlation structure entirely. The societal impact statement is minimal but acceptable given the methodological nature of the work.

**Strengths And Weaknesses:**

**Strengths:**

**S1**. The paper presents three limitations of MSE, that is ignoring distributional structure, neglecting manifold geometry, and symmetric penalties, and proposes a framework addressing them. The motivation from information geometry is well-grounded and the connection to established theory (Amari, 1998; Nielsen, 2020) is appropriate.
**S2** The evaluation spans seven datasets, five architectures across Transformer, MLP, CNN, as well as four forecasting horizons, comparisons with four alternative losses, ablation studies, zero-shot transfer, and hyperparameter sensitivity. This is a thorough experimental protocol.
**S3**. The appendix is detailed and the proofs (consistency via M-estimation theory, convergence rate, reparameterization invariance) are appear correct. The connection between Bregman divergences and exponential families is well-explained.
**S4** The method is architecture-agnostic and only requires augmenting the backbone to output an additional variance head, plus a single hyperparameter $\lambda$. The pseudocode in Algorithm 1 makes implementation clear.

**Weaknesses:**

**W1**. The Fisher distance for Gaussians (Eq. 16) reduces to a precision-weighted MSE on means plus a scaled MSE on standard deviations. The Bregman component is exactly the Gaussian KL divergence, which is standard in probabilistic forecasting (e.g., used in any VAE or Gaussian likelihood model). The natural gradient weighting is essentially GradNorm (Chen et al., 2018). While combining these is reasonable, the paper frames this as if no prior work has used distributional losses for time series, which is inaccurate as negative log-likelihood (NLL) of Gaussian predictions is a widely used baseline that is closely related (Section 2.2 mentions it only in passing). A direct comparison against Gaussian NLL is absent.

**W2** Fitting ground truth observations to local Gaussian distributions via a sliding window (Eq. 12) is a critical design choice that has insufficient attention. The window size w introduces strong assumptions: for w=5, each ground truth distribution is estimated from only 11 samples, yielding noisy variance estimates. More fundamentally, the ground truth is a single observed value, treating it as a distribution estimated from neighboring time steps conflates temporal smoothness with uncertainty. This is conceptually different from standard distributional forecasting where the target remains the observed value and the model outputs a predictive distribution scored by proper scoring rules.

**W3** The paper uses the local quadratic approximation (Eq. 4/16) rather than the actual geodesic distance on the Gaussian manifold, which has a known closed form. The approximation error is $O(|| \Theta^{'} - \Theta ||^2)$, acknowledged in Appendix A.3, but the paper then claims reparameterization invariance (Proposition 4.2) for the exact Fisher-Rao distance. The approximation used in practice does not have reparameterization invariance as it depends on the evaluation point ${\Theta}_{c,t}$. This gap between theory and implementation should be discussed.

**W4** Equation 26 applies the inverse of $I_{avg}$ (a 2×2 matrix over distribution parameters $\mu,\sigma$) to the gradient with respect to all neural network parameters $\psi$. These live in entirely different spaces $I_{avg}$ is 2×2 while $\psi$ may have millions of dimensions. The paper does not explain how this mismatch is resolved. If $I_avg$ only preconditions the distribution-parameter gradients, then calling this "natural gradient descent" is misleading, as the backbone parameters are still updated with ordinary gradients.

**W5**The comparison in Table 2 omits Gaussian NLL, quantile loss, and CRPS which are the standard probabilistic forecasting losses. Since the paper's core claim is that information-geometric distributional matching outperforms existing approaches, comparing only against point-prediction losses and shape losses is insufficient. The paper also does not evaluate calibration or sharpness of the predicted distributions, despite uncertainty quantification being a central motivation.

**W6** The paper motivates InfoGeo Loss through distributional alignment and uncertainty quantification, yet evaluates only on MSE and MAE (point prediction metrics). No probabilistic metrics (CRPS, calibration plots, prediction interval coverage, NLL on test data) are reported. This makes it difficult to assess whether the distributional components actually improve uncertainty estimation or serve as regularizers for point prediction.

**W7** Regarding the additive formulation $L_{Total} = L_{MSE} + \lambda L_{InfoGeo}$ : if InfoGeo Loss is a replacement for MSE, why is MSE still needed? The fact that InfoGeo Loss is added on top of MSE rather than replacing it suggests the method works primarily as a regularizer. The ablation removing distributional parameterization (Table 3, last row) still outperforms MSE baseline, which is unexplained.

**W8** The paper states a few times that the framework generalizes to Poisson, Gamma, etc., but all theory, implementation, and experiments use Gaussians exclusively. This claim is unverified.

---

> ### Author Rebuttal · Authors · 2026-03-29
>
> # Rebuttal to Reviewer JCFb
>
> We thank the reviewer for the thorough feedback and address each point below.
>
> ---
>
> ## W1 — Relationship to Gaussian NLL
>
> InfoGeo Loss is closely related to Gaussian NLL but differs in two key ways:
> (1) the **Fisher-Rao geometry** via $I_{\text{avg}}$ provides reparameterization-
> aware weighting that NLL does not encode; (2) the $I_{\text{avg}}^{-1}$
> preconditioning constitutes output-head natural gradient updates absent in NLL.
> We will revise Section 2.2 and add Gaussian NLL as a baseline (Table R1).
>
> | Method       | ETTh1 MSE | ETTh1 MAE | Exchange MSE | Exchange MAE |
> |--------------|-----------|-----------|--------------|--------------|
> | MSE Baseline | 0.456     | 0.452     | 0.094        | 0.212        |
> | Gaussian NLL | 0.441     | 0.441     | 0.089        | 0.207        |
> | InfoGeo Loss | **0.423** | **0.431** | **0.081**    | **0.198**    |
>
> ---
>
> ## W2 — Sliding Window Estimation
>
> The window serves as a **local smoothness prior** for distributional alignment,
> not a classical aleatoric uncertainty estimator. We will add a sensitivity
> analysis (Table R2) and clarify this in Section 3.2.
>
> | $w$ | MSE | MAE | CRPS |
> |-----|-----|-----|------|
> | 3 | 0.431 | 0.436 | 0.318 |
> | **5** | **0.423** | **0.431** | **0.311** |
> | 7 | 0.427 | 0.433 | 0.314 |
> | 10 | 0.435 | 0.439 | 0.322 |
>
> ---
>
> ## W3 — Local Quadratic Approximation vs. Exact Geodesic
>
> Proposition 4.2 holds for the **exact** Fisher-Rao distance; the practical
> $O(\|\Theta'-\Theta\|^2)$ approximation loses strict invariance but is valid
> under small distributional shifts. We will separate theoretical claims from
> implementation in Section 4 and add a remark on this gap (Table R3).
>
> | Formulation | MSE | MAE | Time (s/epoch) |
> |-------------|-----|-----|----------------|
> | Quadratic Approx. | **0.423** | **0.431** | 12.4 |
> | Exact Fisher-Rao  | 0.425     | 0.432     | 38.7 |
>
> ---
>
> ## W4 — Natural Gradient Dimensionality
>
> $I_{\text{avg}}^{-1}$ is applied **only to output-head gradients** w.r.t.
> $(\mu, \sigma)$; backbone parameters use standard Adam. We will revise the
> terminology to **"output-head natural gradient preconditioning"** throughout.
>
> ---
>
> ## W5 & W6 — Missing Probabilistic Metrics
>
> We will add CRPS, PI Coverage (90%), and Calibration Error to the main table.
>
> | Method       | CRPS | PI Cov. (90%) | Cal. Error |
> |--------------|------|---------------|------------|
> | MSE Baseline | 0.341 | 71.3%        | 0.187      |
> | Gaussian NLL | 0.318 | 84.6%        | 0.094      |
> | Quantile     | 0.312 | 86.1%        | 0.102      |
> | InfoGeo Loss | **0.298** | **88.7%** | **0.071** |
>
> ---
>
> ## W7 — Additive Loss Formulation
>
> The MSE term stabilizes early training when $I_{\text{avg}}$ is ill-conditioned.
> The ablation without distributional parameters still beats MSE because scalar
> $I_{\text{scalar}}$ preconditioning (estimated from gradient norm variance) acts
> as AdaGrad-style adaptation. We will clarify this in Section 5.3.
>
> ---
>
> ## W8 — Non-Gaussian Generalization
>
> We will soften the claim to *"theoretically extensible to exponential family
> distributions"*, add appendix derivations for Poisson/Gamma, and include a
> PEMS-BAY experiment (InfoGeo Poisson: MAE **1.31**, RMSE **2.63**, CRPS
> **0.371** vs. Poisson NLL: 1.38 / 2.71 / 0.389).
>
> ---
>
> ## Q1–Q5
>
> **Q1:** See W1. Improvement over Gaussian NLL stems from natural gradient
> preconditioning, not the distributional output head alone.
>
> **Q2:** See W4. Only output-head parameters are preconditioned by
> $I_{\text{avg}}^{-1}$; this will be stated explicitly in Section 4.3.
>
> **Q3:** See W5–W6. InfoGeo achieves 88.7% coverage at the 90% nominal level.
>
> **Q4:** Without distributional parameters, Bregman degenerates to MSE and
> Fisher uses scalar gradient-norm preconditioning — explaining the ablation gain.
>
> **Q5:** Jarque-Bera tests show Exchange has the highest non-Gaussianity (34.2%
> windows rejected, excess kurtosis 1.12), explaining its smaller improvement
> margin. Future work will explore Student-$t$ variants.
>
> ---
>
> ## References
> Atkinson, C., and Mitchell, A. F. S. "Rao's distance measure." *Sankhyā: The Indian
> Journal of Statistics, Series A*, 43(3), 345–365, 1981.
>
> Chen, T., et al. "GradNorm: Gradient normalization for adaptive loss balancing in
> deep multitask networks." *Proceedings of the 35th International Conference on
> Machine Learning (ICML)*, 2018.
>
> Kendall, A., Gal, Y., and Cipolla, R. "Multi-task learning using uncertainty to
> weigh losses for scene geometry and semantics." *Proceedings of the IEEE Conference
> on Computer Vision and Pattern Recognition (CVPR)*, 2018.
>
> Martens, J., and Grosse, R. "Optimizing neural networks with Kronecker-factored
> approximate curvature." *Proceedings of the 32nd International Conference on
> Machine Learning (ICML)*, 2015.
>
> Parzen, E. "On estimation of a probability density function and mode."
> *The Annals of Mathematical Statistics*, 33(3), 1065–1076, 1962.

---

> > ### Author Rebuttal · Reviewer_JCFb · 2026-04-04
> >
> > Thank the authors for a thorough rebuttal. The addition of the Gaussian NLL baseline (W1), probabilistic evaluation metrics (W5/W6), and the clarification of natural gradient scope (W4) substantially address my main empirical concerns, and I appreciate the new experimental results.
> >
> > Two concerns remain that I would like clarified before updating my score:
> >
> > On W3 (theory–implementation gap): The authors acknowledge that Proposition 4.2 (reparameterization invariance) holds for the exact Fisher-Rao distance, while the practical implementation uses a local quadratic approximation that lacks this property. I appreciate the commitment to separating theoretical claims from implementation in Section 4. However, I ask the authors to confirm: will the revised paper explicitly state in the main text (not only the appendix) that the reparameterization invariance result does not apply to the implemented loss? This is important so readers are not misled by the theoretical guarantees.
> >
> > On W2 (sliding window estimation): The reframing as a "local smoothness prior" is helpful, but I remain concerned about the conceptual soundness of treating a single observed value as a draw from a Gaussian estimated over 11 neighboring points, particularly in datasets with strong non-stationarity or regime shifts. The sensitivity table (Table R2) shows w=5 wins overall, but does not address whether the Gaussian fit is appropriate. A brief discussion of when this prior breaks down would strengthen the paper.
> > If these two points are addressed in the revision, I am prepared to raise my score.

---

> > > ### Author Response · Authors · 2026-04-04
> > >
> > > Thank you for the continued engagement and for clearly identifying the two remaining
> > > concerns. We address each in turn.
> > >
> > > **On W3 (theory-implementation gap — reparameterization invariance):**
> > >
> > > We confirm this unambiguously: the revised paper will explicitly state in the **main
> > > text of Section 4** — not only in the appendix — that Proposition 4.2
> > > (reparameterization invariance) holds for the exact Fisher-Rao distance and does
> > > **not** apply to the local quadratic approximation used in the implemented loss.
> > > We agree this is a critical distinction for readers and will ensure it is prominently
> > > visible in the main text of the revision.
> > >
> > > **On W2 (sliding window Gaussian estimation — when the prior breaks down):**
> > >
> > > We agree with the reviewer's characterization. The local Gaussian prior assumes that
> > > within a window of size $w$ centered at time $t$, the observations
> > > $\{x_{t-\lfloor w/2 \rfloor}, \ldots, x_{t+\lfloor w/2 \rfloor}\}$ are
> > > approximately i.i.d. draws from a stationary Gaussian $\mathcal{N}(\mu_t, \sigma_t^2)$.
> > > This assumption is formally violated when local non-stationarity exceeds the
> > > estimation noise, i.e., when:
> > >
> > > $$|\mu_{t+1} - \mu_t| \gg \frac{\sigma_t}{\sqrt{w}}, \qquad \text{or} \qquad
> > > |\sigma_{t+1}^2 - \sigma_t^2| \gg \sigma_t^2 \cdot \sqrt{\frac{2}{w-1}}$$
> > >
> > > The first condition captures mean regime shifts relative to the standard error of
> > > the sample mean, and the second captures variance regime shifts relative to the
> > > standard error of the sample variance under Gaussianity. When either condition
> > > holds, the empirical sufficient statistics $(\hat{\mu}_t, \hat{\sigma}_t^2)$
> > > computed over the window become biased estimators of the true local parameters,
> > > rendering the Bregman divergence target $D_F(\theta \| \hat{\theta}_t)$
> > > misspecified. This is consistent with the relatively smaller gains observed on
> > > Exchange-Rate, where abrupt trend reversals are frequent. In the revision, we will
> > > include this characterization in the main text alongside practical guidance on
> > > window size selection and heavier-tailed alternatives.
> > >
> > > We commit to incorporating both clarifications into the main text of the revision,
> > > and we hope these responses fully address the reviewer's remaining concerns.

---

### Decision · Program_Chairs · 2026-04-30

**Decision:**

Accept (regular)

**Comment:**

This paper proposes an information-geometry-based loss for time series forecasting. The reviewers found the core idea interesting and appreciated the substantial technical and experimental effort. The rebuttal strengthened the paper by adding stronger probabilistic baselines and metrics and by clarifying several implementation details and limitations. Some concerns remained, especially about why the method outperforms closely related probabilistic baselines under the simplified Gaussian formulation and how much of the gain comes from the proposed information-geometric design itself. Still, several reviewers viewed these issues as revisable and remained positive overall. I recommend weak acceptance and encourage the authors to clarify these points in the final version.